# Prediction and Process Analysis of Tensile Properties of Sinter-Hardened Alloy Steel by Artificial Neural Network

**Zhaoqiang Tan [1,2], Zijun Qin [1], Qing Zhang [2], Yong Liu [1] and Feng Liu [1,*]**

[1] State Key Laboratory of Powder Metallurgy, Central South University, Changsha 410083, China; simon.tan@hoganas.com (Z.T.); zijun.qin@csu.edu.cn (Z.Q.); yonliu@csu.edu.cn (Y.L.)

[2] Höganäs China Co., Ltd., Shanghai 201799, China; sunny.zhang@hoganas.com

[*] Correspondence: liufeng@csu.edu.cn

**Abstract:** Sinter-hardening is an emerging powder metallurgy process by which the consolidation of powder and the hardening of dense bulk samples are integrated into one step. In this study, to understand the complex effects of sinter-hardening parameters on the properties of the Fe-Cr-Ni (Cu)-C alloy, an artificial neural network (ANN) with the topology of a nonlinear multi-layered perceptron was designed to predict the ultimate tensile strength and elongation, considering parameters including chemical composition, sintering temperature, and cooling rate. The predictability of the ANN was verified by experiments, indicating that this method is adequate to quantitatively ascribe steel properties to powder metallurgy parameters in the view of improving process robustness.

**Keywords:** powder metallurgy; tensile properties; sinter-hardening; artificial neural network





## 1. Introduction

Powder metallurgy (PM) is a net-shape and cost-effective mass production route of components with satisfactory mechanical properties [1]. As a novel PM process free of secondary quenching, sinter-hardening has a high efficiency of consolidation and excellent size accuracy; hence, it can be applied to produce high-performance PM alloy steel. By rapid cooling, the powders consolidated at sintering temperatures with cold inert gas, and martensite was readily formed in the steel matrix of the bulk sample during sinter-hardening [2,3].

Traditionally, copper, phosphorus, nickel, and molybdenum were widely used as alloying elements in powder metallurgy, generating a comprehensive materials system from low to high strength. In industrial practice, the carbon and low-alloy Ni-, Mo-, and Cu-alloyed steel are sintered in a continuous belt furnace at a temperature between 1100 and 1180 °C, since the resulting mechanical properties are high enough to match the technical requirements of many structural parts used in different applications and, in particular, in automotive industry. Today, more studies on cost-efficient alloy elements such as Cr, Mn, and Si are carried out. Cr and Mn alloying elements provide excellent hardenability and strengthening effects while maintaining cost effectiveness [4–6]. However, Cr and Mn elements are prone to oxidization. By utilizing pre-alloyed Cr containing powders in combination with high-temperature sintering, namely > 1200 °C, a significantly improved strength can be achieved [7]. A study also reviewed the factors of density, cooling rate, and sintering temperature related to the sinter-hardening process on the performance [8]. A high sintering temperature for sintered steel with Ni elements enhances the compositional homogeneity of the metallic matrix, decreases the quantity of Ni-rich austenite, and finally, enhances mechanical properties.

The interest of Fe-Cr materials for sintering–hardening applications is growing. However, during the design stage of a PM component using the sintering–hardening process, engineers are still suffering from the lack of suitable and reliable guidelines for the selection of appropriate chemical compositions and process parameters. A good prediction model

will be important in helping designers with material selection, verification of properties attainable through the process, cost implications, etc.

To optimize the mechanical properties manufactured by sinter-hardening, it is critical to understand how the alloying elements, sintering temperature, and cooling rate affect the obtained microstructure. Previous work indicated that the homogenization of alloying elements at high sintering temperatures led to the coarsening of austenitic grains, which weakened the alloy strength [9]. However, the homogenization of elements including Cr, Cu, Ni and C is necessary to strengthen the matrix. Thereby, the interactions among elements and the processing are complex and major concerns when looking to obtain the desired properties.

The use of the traditional "trial and error" method is time- and cost-consuming to solve the above-mentioned issue. To address that, data modeling by an artificial neural network (ANN) could build a nonlinear correlation integrating consolidation parameters with performance that enables an associative memory analogous to that of powder metallurgy experts [10]. Actually, ANNs have already been extensively used in the prediction and simulation of complex problems in material science. Numerous studies have been published on the prediction of mechanical properties and microstructures depending on the elemental composition and processing conditions. Benchmark ANN models and multilinear regression models have been developed to predict the tensile strength of low-carbon bainitic steel, and a much higher accuracy was exhibited by the ANN model [11]. A three-layer ANN model was proposed in predicting the fatigue crack growth and applied to the material of aluminum alloy, and the results show a higher accuracy than commonly used Walker and Forman models [12]. In PM applications, there is also more and more research on this new branch of powerful and effective computing tools. For example, soft computation was attempted for the data modeling of the powder metallurgy process [13,14]. The sintering temperature of pure iron was optimized by the ANN to achieve desired stress vs. strain behavior [15]. Fatigue strength was predicted by a multi-layer feed-forward back-propagation ANN for porous Fe-Cu-C steel after sintering, and consolidation parameters were optimized by a genetic algorithm, guaranteeing satisfactory fatigue strength [16]. In addition to performance prediction, ANN is able to solve an inverse problem, which refers to determining consolidation parameters as per the requirement and/or specification of the mechanical performance. For instance, a model combining an ANN and rule-based inference was established to recommend a powder metallurgy scheme [17].

In this study, an ANN with a topology of multi-layered perceptron learning with a back-propagation algorithm was employed to predict the ultimate tensile strength and elongation of sinter-hardened Fe-Cr- Ni (Cu)-C alloy steel. The data modeling performed by the perceptron quantitatively elucidates the correlation of tensile properties to consolidation parameters. Experimental data were collected not only for the training but also the calibration of the perceptron, which verified the effect of data modeling with the accuracy criterion. Jominy hardenability testing and microstructure characterization were carried out to rationalize the achievement of the data modeling by ANN.

## 2. Materials and Methods

For the data modeling by ANN, 168 pairs of experimental data (obtained from a Höganäs database) listed in supplementary files were collected, of which 151 pairs were used for training, and the remaining 17 pairs were selected to calibrate a multi-layered perceptron aimed at the analysis of ultimate tensile strength and elongation in consideration of parameters such as alloying composition, sintering temperature, and cooling rate. Table 1 lists the variation and varying rates of the parameters for the prediction of tensile properties by the ANN.

**Table 1.** Predictive range of the parameters for predicting tensile performance by ANN.

| Composition and Technology | Cu (wt.%) | Ni (wt.%) | C (wt.%) | Sintering Temperature (°C) | Cooling Rate (°C s$^{-1}$) |
|---|---|---|---|---|---|
| Range | 0~1 | 1~2 | 0~1 | 1120–1250 | 1~8 |
| Step length | 0.2 | 0.5 | 0.2 | 1120, 1200, 1250 | 0.5 |

The input data in this work were generated in the tech center of Höganäs China Co., Ltd (Shanghai, China). Blended with Ni, Cu, graphite, and 0.6 wt.% lubricant, water-atomized pre-alloyed Fe-Cr powders with ~1.8 wt.%Cr (Höganäs AB, Höganäs, Sweden) were compacted with a green density of 7.0 g/cm$^3$. The compact was sintered at 1120~1250 °C for 30 min in an atmosphere of the mixture of 10 vol.% $H_2$ and 90 vol.% $N_2$ in a commercial sinter-hardening furnace with a cooling rate of 0.5–8 °C/s. The furnace was driven by the pushing system in the dewaxing zone and a rolling system in the sintering zone and cooling zone separately. The sinter-hardened steel was tempered at 180 °C for 60 min in air. Tensile testing was conducted with the Zwick/Roell Z100 (ZwickRoll GmbH & Co., KG, Ulm, Germany) universal testing machine following the standard ISO2740:2009. To study the hardenability of the materials, some cylindrical samples with a diameter of 6 mm and a height of 10 mm were pressed to 7.0 g/cm$^3$ and sintered in a 90$N_2$/10$H_2$ atmosphere for 20 min at 1120 °C. They were then held at 960 °C in vacuum in the dilatometer for 10 min before cooling at rates of 0.1–10 °C/s in a helium atmosphere. Continuous cooling transformation (CCT) diagrams were depicted by quench dilatometry equipped with its own software. Phase transformation points were identified from the CCT curve. To study the microstructure and explain the behavior of the properties, metallographic samples were chopped and polished along the cross-section of the training tested bars. The microstructure of the sinter-hardened steel was inspected by optical microscopy.

## 3. Results and Discussion

### 3.1. Architecture of the Multi-Layered Perceptron

With a nonlinear multi-layered perceptron, ANN data modeling was conducted for Fe-Cr alloy steel in order to correlate consolidation parameters with tensile properties. To be exact, a quantitative analysis was provided to correlate the contents of Ni, Cu, and C, sintering temperature, and cooling rate to ultimate tensile strength and elongation. First, a preliminary z-normalization was handled with the input data of ANN, as given by

$$z = \frac{x - \overline{x}}{S} \qquad (1)$$

where $x$ is the original value of an input datum, $\overline{x}$ is the mean value, and $S$ is the standard deviation of such data. From a total 168 pairs of data, 151 pairs were used in the training dataset, among which 1/10 data values were randomly taken to fix hyperparameters, such as the number of layers, number of nodes, number of epochs, and so on. The remaining 17 pairs showed in Table 2 were adopted to calibrate the model. The multi-layered perceptron was trained by a supervised error back-propagation algorithm, with which the weights linking neurons between layers were updated for minimizing an error ($E$) given by

$$E = \frac{1}{N} \sum_{i=1}^{N} (p_i - d_i)^2 \qquad (2)$$

where $p_i$ and $d_i$ are the computed and desired output, and $N$ (=151) is the number of the pairs of data used for training, respectively. The trained perceptron was calibrated with 17 pairs of data with an accuracy constrained by the mean squared error ($MSE$) and the coefficient of consistency ($R^2$). $MSE$ is given by

$$MSE = \frac{1}{n} \sum_{i=1}^{n} (y_i - \hat{y}_i)^2 \qquad (3)$$

where $n$ (=17) is the number of data pairs for the calibration, and $y_i$ and $\hat{y}_i$ are the actual (target) value and the value predicted by the perceptron, respectively. $R^2$ measures the consistency between the predictions by ANN and the data used for calibration, namely,

$$R^2 = 1 - \frac{\sum\limits_{i=1}^{n}(y_i - \hat{y}_i)^2}{\sum\limits_{i=1}^{n}(\overline{y}_i - y_i)^2} \tag{4}$$

where $\overline{y}_i$ is the mean value of $y_i$.

**Table 2.** Consolidation parameters and experimental values used for calibration.

| | Consolidation Parameters | | | | | Experimental Values | |
|---|---|---|---|---|---|---|---|
| Exp. no | Cu (wt.%) | Ni (wt.%) | C (wt.%) | Sintering T (°C) | Cooling Rate (°C s$^{-1}$) | Tensile Strength (MPa) | Elongation (MPa) |
| 1 | 1 | 0 | 0.6 | 1120 | 3 | 903 | 0.37 |
| 2 | 1 | 0 | 0.6 | 1250 | 3.5 | 1035 | 1.28 |
| 3 | 0 | 0 | 0.6 | 1250 | 3 | 773 | 2.54 |
| 4 | 1 | 0 | 0.6 | 1250 | 6 | 941 | 0.10 |
| 5 | 0 | 2 | 0.4 | 1120 | 6 | 680 | 1.62 |
| 6 | 1 | 0 | 0.6 | 1200 | 6 | 1009 | 0.20 |
| 7 | 0 | 2 | 0.6 | 1120 | 3 | 957 | 0.45 |
| 8 | 0 | 2 | 0.6 | 1120 | 2 | 935 | 1.14 |
| 9 | 0 | 0 | 0.6 | 1120 | 1 | 703 | 1.40 |
| 10 | 0 | 2 | 0.6 | 1250 | 3 | 1082 | 0.47 |
| 11 | 1 | 0 | 0.6 | 1250 | 3 | 1042 | 0.97 |
| 12 | 0 | 0 | 0.6 | 1250 | 1 | 840 | 2.21 |
| 13 | 0 | 2 | 0 | 1120 | 1 | 295 | 2.55 |
| 14 | 0 | 2 | 0.4 | 1120 | 8 | 677 | 1.26 |
| 15 | 0 | 0 | 0.8 | 1200 | 1 | 850 | 2.18 |
| 16 | 1 | 0 | 0.6 | 1120 | 6 | 919 | 0.21 |
| 17 | 0 | 0 | 0.8 | 1200 | 1 | 844 | 2.27 |

The architecture of the perceptron and the convergence of its training are shown in Figure 1. The nonlinear multi-layered perceptron has a topology of feed-forward ANN comprising three categories of layers of neurons, i.e., the input, the hidden, and the output. The hidden layer is composed of three layers, i.e., two dense and one dropout. The neurons in all the layers worked as a rectified linear unit, except the one in the dropout layer, the latter of which was assigned with a nonlinear activation function. The perceptron for either ultimate tensile strength or elongation was in convergence after an iteration of training for 500 epochs. The MSE and $R^2$ of the tensile strength prediction model and elongation prediction model are both 0.01 and 0.89.

A total of 17 pairs of actual data points listed in Table 2 generated at various consolidation parameters and alloying compositions and broader results from low to high strength were selected randomly to validate the model. Figure 2a,c compare the actual values with the ones predicted by the perceptron. The green symbol denotes the data for training. The purple delta represents the data of validation. The yellow line is the baseline that implies 100% consistency. All the data, whether for training or for validation, are uniformly concentrated on the baseline, which signifies the adequate capability of property prediction by the ANN. Figure 2b,d show the prediction by ANN with the data of validation. For more verification of the reliability of the trained neural network, the average relative error of the validation dataset was also calculated to be 6.87% for tensile strength and 0.25% for elongation, which proves that the network successfully modeled the input data with an acceptable margin of error. Obviously, the data predicted by the perceptron are in good agreement with the actual property data of alloy steel.

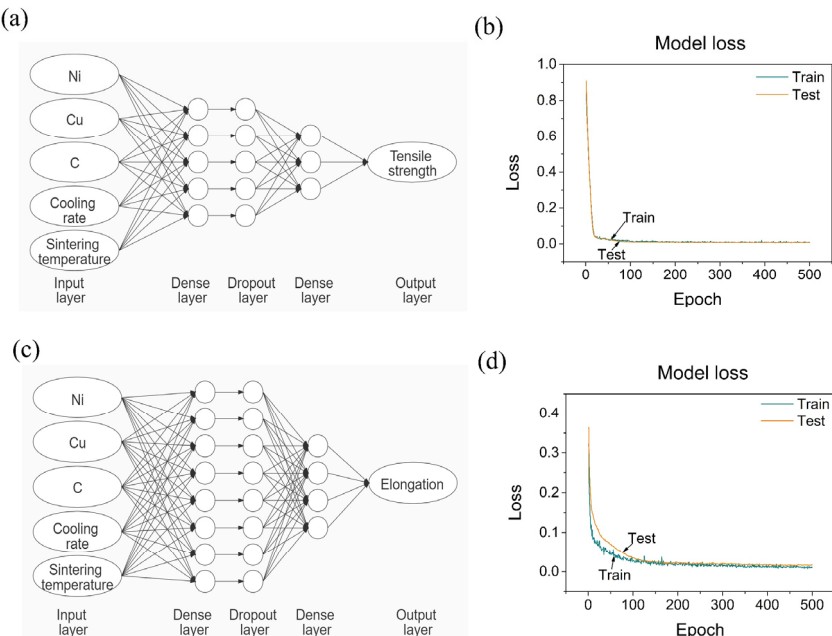

**Figure 1.** The architecture and convergence curves of the multi-layered perceptron: (**a**) architecture and (**b**) convergence for ultimate tensile strength; (**c**) architecture and (**d**) convergence for elongation.

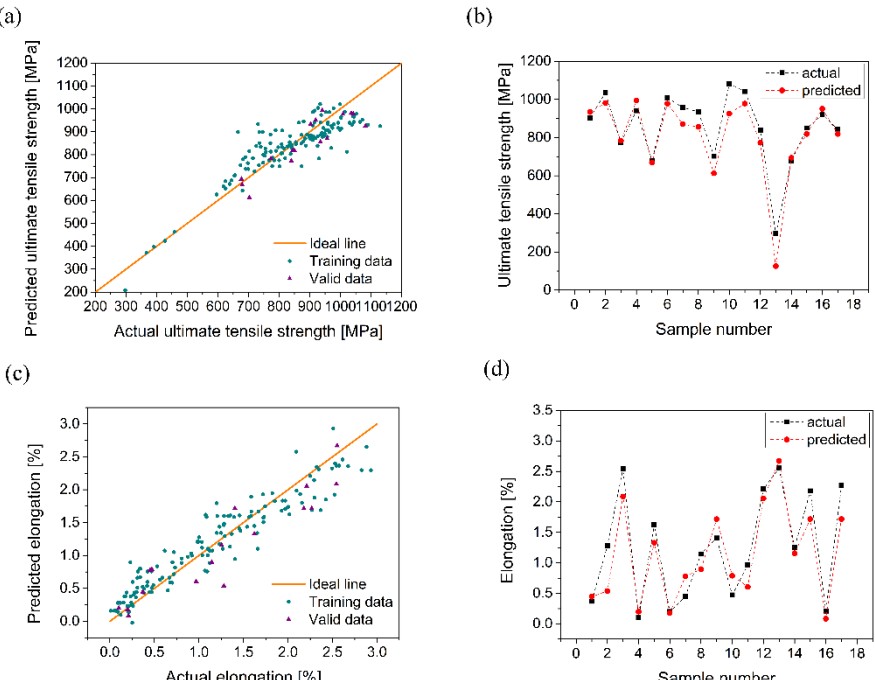

**Figure 2.** The prediction by ANN: the comparison between actual and predicted values of (**a**) ultimate tensile strength prediction model, and (**b**) the comparison in validation dataset; (**c**) comparison between the actual and predicted value of elongation prediction model, and (**d**) the comparison in validation dataset.

### 3.2. Prediction of Tensile Properties by Perceptron

The complexity of the sinter-hardening process with alloying compositions and consolidation parameters (e.g., sintering temperature and cooling rate) is responsible for the difficulty to predict tensile properties by ANN. The dataset of tensile performance with 14,850 values was successfully predicted and visualized for the alloys with various compo-

sitions and consolidation parameters. As a result, these predicted data can be employed inversely to further analyze the process and relationship among the neurons.

Taking the samples that sintered at 1120 °C as an instance, Figure 3a shows the prediction of tensile performance by ANN as influenced by carbon content and cooling rate. The increase in carbon content implies the improvement in ultimate tensile strength; this happens more significantly at an inferior level of carbon content (0~0.6 wt.%), at which ultimate tensile strength is insensitive to cooling rate. It is worth noting that consolidation at a moderate and high cooling rate, or higher carbon content (0.6–0.8 wt.%), could not contribute much further to the ultimate tensile strength. As shown in Figure 3b,c, ultimate tensile strength was considerably improved with the Ni or Cu content and cooling rate in alloy steel (~0.6 wt.% carbon) sintered at 1120 °C until both consolidation parameters were increased to 1.5 wt.% (for Ni) or 2.0 wt.% (for Cu) and 3 °C/s, respectively.

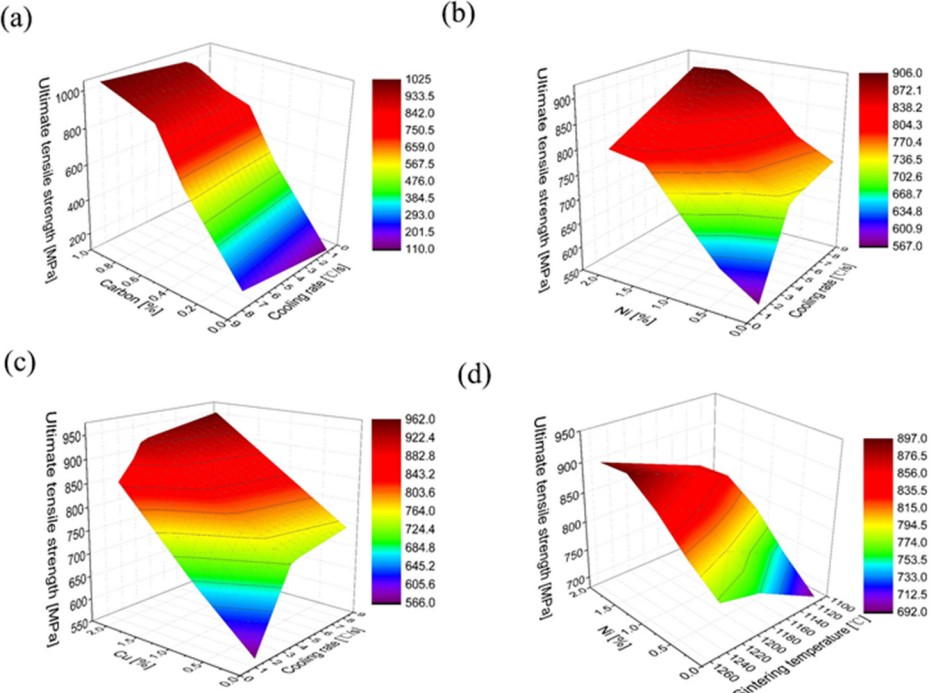

**Figure 3.** Effects of alloy compositions and consolidation parameters on the ultimate tensile strength: (**a**) effects of carbon and cooling rate on ultimate tensile strength of Fe-1.8Cr-2Ni sintered at 1120 °C, (**b**) effects of Ni content, (**c**) Cu content, and cooling rate on ultimate tensile strength of Fe-1.8Cr-0.6C sintered at 1120 °C, (**d**) effects of Ni content and sintering temperature on ultimate tensile strength of sinter-hardened Fe-1.8Cr-0.6C cooled at 2.5 °C/s.

As shown in Figure 4, the CCT diagram of Fe-1.8Cr and Fe-1.8Cr-Cu/Ni is depicted with a carbon content of 0.6 wt.%. The nucleation of ferrite was prohibited with the addition of Cu or Ni, with which austenite ($\gamma$ phase) was stabilized. In fact, Cu augments the $\gamma$-field, and Ni opens the $\gamma$-field [18]. The pearlite and bainite transformation curves move toward the region, representing a slower cooling rate. As shown in Figure 5a,b, for Fe-1.8Cr-0.6C, a fully pearlite phase is presented at a cooling rate of 3 °C/s. With the addition of 1 wt.% Cu and 2 wt.% Ni, a martensitic-based structure with some pearlite and bainite islands was formed by cooling at 3 °C/s. During sinter-hardening, as much martensite as possible is desired for improving the hardenability of the sintered material. There are two types of martensite, i.e., brittle plate and ductile lath. As a result, micro-cracks were likely to form in plate martensite, having a negative effect on ultimate tensile strength of high-carbon steel [19,20]. However, lath martensite is prone to be produced in low-carbon steels, as shown in Figure 5c. It is desirable to form hybrid microstructures mixing both types of martensite or even pure plate martensite (as shown in Figure 5d), which is practically

achievable with the addition of 0.6~1.0 wt.% carbon. With an increase in the cooling rate for Fe-1.8Cr-2Ni (in wt.%), ultimate tensile strength was improved with plenty of lath martensite readily formed by adding carbon less than 0.6 wt.% but impaired with more plate martensite formed with higher levels of alloy content. This is the reason why high carbon content and a higher cooling rate cannot further improve the tensile strength.

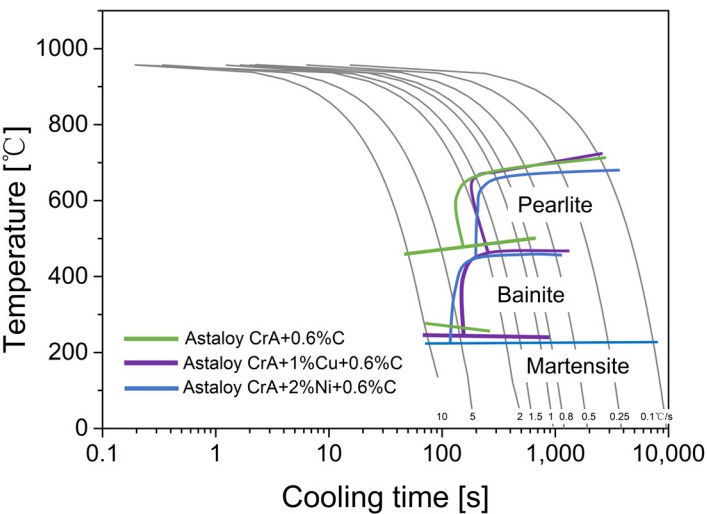

**Figure 4.** CCT diagram of the alloy system depicted by the Jominy hardenability test.

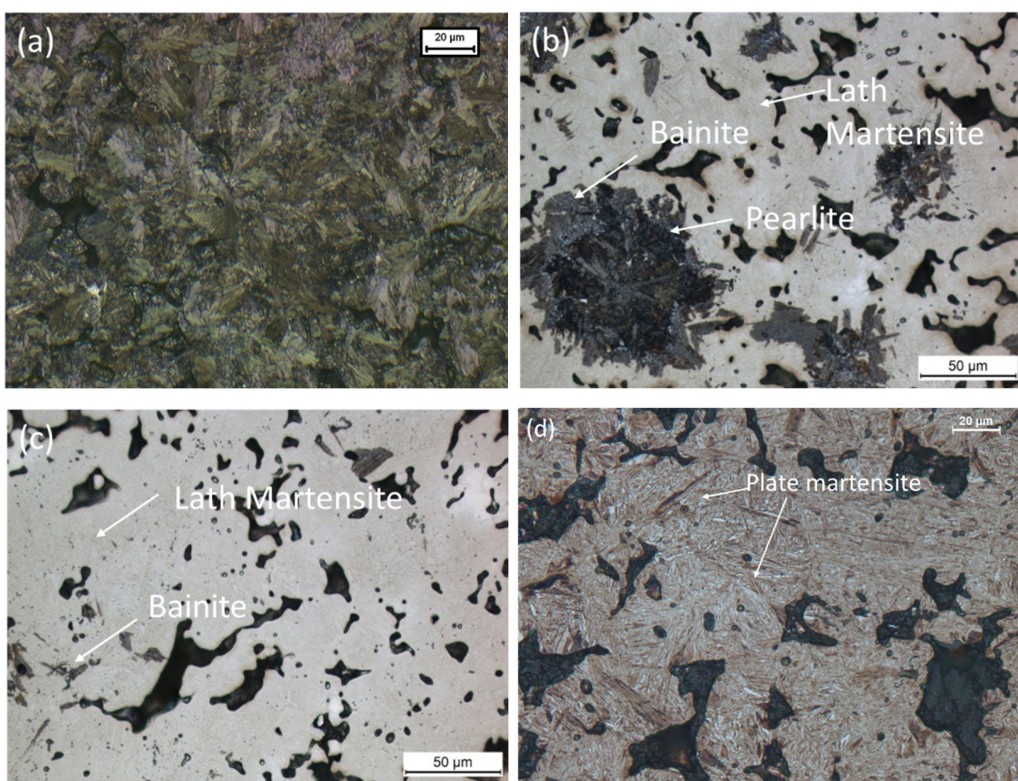

**Figure 5.** Phases constitution of materials: (**a**) Fe-1.8Cr-0.6C and (**b**) Fe-1.8Cr-2Ni-0.6C sintered at 1120 °C and cooled at 2.5 °C/s, (**c**) Fe-1.8Cr-2Ni-0.6C and (**d**) Fe-1.8Cr-2Ni-0.8C sintered at 1120 °C and cooled at 5° C/s.

Sintering temperature is another key factor for the sinter-hardening process. Figure 3d shows a response surface plot correlating ultimate tensile strength to Ni content and sintering temperature. The increase in sintering temperature from 1120 to 1250 °C benefited

higher tensile performance for Fe-1.8Cr blended with Ni. This result is consistent with results reported elsewhere [21–23]. Moreover, the formation of sintering necks and the roundness of pores were promoted at high-temperature sintering. The substantial formation of round pores (free of acute angles) is beneficial to the improvement of ultimate tensile strength because neither the origination nor the propagation of cracks sensitively occurs in the vicinity of round pores under the load by tension. Furthermore, for an Fe-Cr system, a higher level of sintering temperature could ensure good feasibility of the reduction in Cr-containing oxides [24].

Ultimate tensile strength remained constant once the content of nickel was above ~1.5 wt.%. In this study, Ni was admixed as an enhancer of the hardenability of powders. The hardening effect of alloying elements played a part in the solid solution. At an inferior level of sintering temperature, the distribution of Ni is heterogeneous, whereby the austenite enriched with Ni is likely to be formed at this site. However, the austenite formed might be fragile, acting as a crack origination site, and be responsible for the ultimate failure under tension [25]. Austenite enriched with Ni is observable at various sintering temperatures, although the concentration of Ni in austenite varies with sintering temperature. A smaller area of Ni-rich austenite and bainite is viable at higher temperatures, which indicates that a more homogeneous distribution of Ni is viable, as clearly evidenced in Figure 6, showing the distribution of nickel in Fe-1.8Cr sintered at 1120 °C, 1200 °C, and 1250 °C, respectively. In this case, the sufficient diffusion of Ni into the Fe-1.8Cr matrix induced significant, effective hardenability. Thus, plenty of martensite could be produced by sintering at higher temperatures. Due to great hardenability with sufficient Ni dissolution, plate martensite was more prone to be formed at higher temperatures.

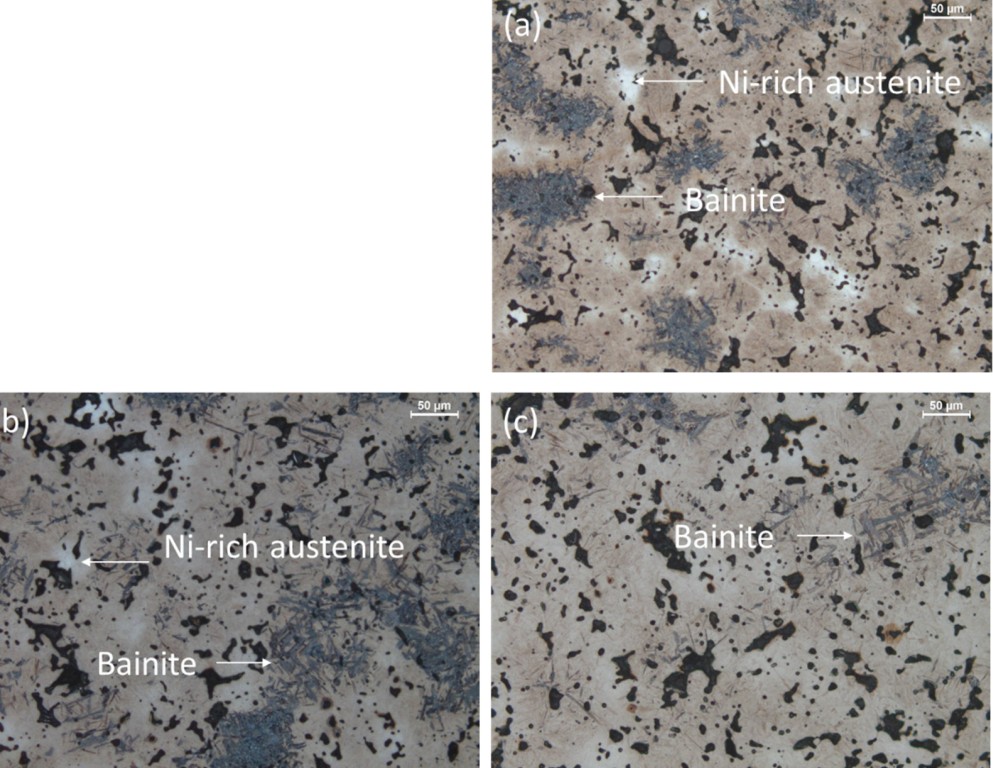

**Figure 6.** The microstructures of alloying elements for Fe-1.8Cr-2Ni-0.6C sintered at (**a**) 1120 °C, (**b**) 1200 °C, and (**c**) 1250 °C.

The nonlinear correlation of alloying content and consolidation parameters to tensile strength predicted by the ANN model was rational, as proven by the microstructure and hardenability of material Fe-Cr-Ni (Cu)-C, and the optimization of consolidation parameters can be guided when alloy compositions are designed to achieve the best tensile strength.

Thus, the ANN prediction model was proven to be reliable as a powerful tool for the development of powder metallurgy components, since it facilitates material composition selection and processing conditions determination, which will contribute to significant lead-time and cost reductions.

## 4. Conclusions

An ANN, in the form of a nonlinear multi-layered perceptron, was designed to predict the patterns correlating alloying elements (of Cu, Ni, and C) and consolidation parameters (sintering temperature and cooling rate) to tensile properties (ultimate tensile strength and elongation) of sinter-hardened Fe-Cr-Ni(Cu)-C alloy steel. The excellent agreement of data modeling by the ANN with the experimental data was demonstrated, with the MSE and $R^2$ of the tensile strength prediction model and the elongation prediction model being 0.01 and 0.89, respectively. Data modeling by ANN is adequate to quantitatively ascribe steel properties to powder metallurgy parameters, showing that the nonlinear relationship of tensile strength is affected by alloy content and process parameters. Moreover, it was explained that the impairment of tensile strength of Fe-Cr-Ni(Cu)-C predicted by the model was the result of plate martensite formed in the conditions of higher alloying content, a higher cooling rate, or more diffusion of Ni (Cu) at high sintering temperatures.

**Author Contributions:** Conceptualization methodology, Z.T., Q.Z.; software., Z.Q.; validation, Z.T., Q.Z., Z.Q., F.L.; investigation, Z.T.; data curation, Z.T.; writing—original draft preparation, Z.T., Q.Z.; writing—review and editing, Z.T., F.L.; supervision, F.L., Y.L.; project administration, Y.L. All authors have read and agreed to the published version of the manuscript.

**Funding:** This research received no external funding.

**Institutional Review Board Statement:** Not applicable.

**Informed Consent Statement:** Not applicable.

**Data Availability Statement:** Data are contained within the article.

**Acknowledgments:** Höganäs China Co., Ltd. is gratefully acknowledged for providing experimental equipment and materials.

**Conflicts of Interest:** The authors declare no conflict of interest.

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
