# Peer review of "Prediction and Process Analysis of Tensile Properties of Sinter-Hardened Alloy Steel by Artificial Neural Network"

_metals, doi:10.3390/met12030381_

Round 1

Reviewer 1 Report

Prediction of the mechanical properties of the new industrial materials using Artificial Neural Network (ANN) is a very important research subject which results are extremely important to change the future of materials science.

A significant reduction of the number and amounts of materials used in research before the solutions are brought to the market is needed especially in the COVID pandemic times when raw material supply chains are disrupted or significantly delayed.

The manuscript entitled "Prediction and Process Analysis of Tensile Properties of Sinter-hardened Alloy Steel by Artificial Neural Network" undoubtedly refers to the important artificial intelligence/machine learning field of research however it cannot be accepted in the current form.

First of all, the reader is very confused about the idea of the experiment and the key findings and conclusions presented by the authors. Before publication, the materials and methods section must be significantly improved, because in the current form reader cannot find results of the empirical experiment performed based on the ANN prediction. The empirical experiment results supported with materials microstructures should be presented and explained to the reader in the discussion section.

I would suggest describing broader in the materials and method section and/or adding supplementary data (for example table) with 168 "pairs" of experimental data used for training and calibration. The authors of the manuscript should mark both data used for training and data used for calibration. Furthermore, it is important here to describe the range of used alloy alloying elements (Cu, Ni, C) and experimental results (tensile and elongation) depending on the cooling rates.

Figure 3 shows the prediction of 14.850 experimental results obtained by ANN with a variation of the parameters summarized in Table 1. However, figure 3 quality is very poor, and it is very debatable to agree that discussion about the influence of carbon level (0.65) on tensile properties is accurate.

Furthermore, it should be shown in the manuscript that ANN was accurate by the presentation of the verification of the ANN by the new experimental empirical results of mechanical properties of Fe-Cr-Cu (Ni)-C alloy with compositions not used as training data. The new experimental empirical data should be presented on the background of prediction and training data - the same figure. If the "valid data" in Fig 2 is the empirical data which is verification of ANN results (tensile and elongation from a few alloys) it should be described in the results section. Why results in Fig 2. b) and d) are connected in one line? Where is information about fabrication details and alloy composition of these 18 samples? Where is shown their microstructure?

Moreover, the discussion of the microstructure has no sense without figures with microstructure and marked phases. The presented in the discussion SEM and EDS are unacceptable (extremely low quality) and have no connection to the performed predicted experiments. In the prediction, model authors have not used any microstructure data so the discussion should focus on the tensile properties obtained experimentally and their comparison to ANN prediction results. The material microstructure has of course main influence on tensile properties, but some microstructure results must be present to perform discussion.

Last but not least the authors should add some information about  Fe-Cr-Cu (Ni)-C alloy applications and why this alloy was chosen to perform the study.

Reviewer 2 Report

The authors have studied the tensile properties of sinter hardened Fe-Cr-Cu (Ni)-C alloy steel by experiment and artificial neural network verification. This work is interesting and can be used for developing alloy steels in steel industries. However, there are several issues in this manuscript.

  1. The presentation of characterization results is very poor. For example, Figure 4 shows a martensite plate. The other features are not labeled well. The authors say there is no precipitation but it is not convincing from this image.
  2. The location of scale bars is ambiguous and hence difficult to comment on the results.
  3. The fonts in Fig. 1 should be enhanced for better clarity.
  4. The clarity of Fig. 3 is poor.
  5. Additionally, the language should be polished.

Round 2

Reviewer 1 Report

The authors used the same data for validation and testing what is unacceptable in machine learning in materials science.  

Authors should use natural network setup solutions (dividing data for 1. training, 2. cross-validation, 3. testing) that are already published for example in https://doi.org/10.1016/j.mtcomm.2021.102022.

Furthermore,  the introduction should explain how the presented model is different from models available nowadays https://www.nature.com/articles/s41578-021-00340-w.

I would suggest also improving metallography in the presented manuscripts. The reader would like to see grain boundaries in the materials https://www.mdpi.com/2076-3417/11/17/7951/htm.

Reviewer 2 Report

The authors have addressed all my comments. 

Author Response

Thanks for reviewer's comment. Your suggestions are quite valuable and helpful to finalize the paper. 

Round 3

Reviewer 1 Report

Thank you for your answer and explanations. However, before publication, I suggest adding in the introduction a paragraph about the prediction of the properties of the metallic materials based on the ANN protocols using at least a few reference documents. You have used a similar protocol to other authors so you should at least comment on their results.
